# Development of a COVID-19 Vulnerability Index (CVI) for the Counties and Residents of New Jersey, USA

**DOI:** 10.3390/ijerph20136312

**Published:** 2023-07-07

**Authors:** Remo DiSalvatore, Sarah K. Bauer, Jeong Eun Ahn, Kauser Jahan

**Affiliations:** 1Department of Civil and Environmental Engineering, Rowan University, Glassboro, NJ 08028, USA; rdisalvatore3@gmail.com (R.D.); jahan@rowan.edu (K.J.); 2Department of Environmental and Civil Engineering, Mercer University, Macon, GA 31207, USA; bauer_sk@mercer.edu

**Keywords:** COVID-19, index, virology, pandemic, vulnerability, public health

## Abstract

The coronavirus disease 2019, or COVID-19, has impacted countless aspects of everyday life since it was declared a global pandemic by the World Health Organization in March of 2020. From societal to economic impacts, COVID-19 and its variants will leave a lasting impact on our society and the world. During the height of the pandemic, it became increasingly evident that indices, such as the Center for Disease Control’s (CDC) Social Vulnerability Index (SVI), were instrumental in predicting vulnerabilities within a community. The CDC’s SVI provides important estimates on which communities will be more susceptible to ‘hazard events’ by compiling a variety of data from the U.S. Census and the American Community Survey. The CDC’s SVI does not directly consider the susceptibility of a community to a global pandemic, such as the COVID-19 pandemic, due to the four themes and 15 factors that contribute to the index. Thus, the objective of this research is to develop a COVID-19 Vulnerability Index, or CVI, to evaluate a community’s susceptibility to future pandemics. With 15 factors considered for CDC’s SVI, 26 other factors were also considered for the development of the CVI that covered themes such as socioeconomic status, environmental factors, healthcare capacity, epidemiological factors, and disability. All factors were equally weighted to calculate the CVI based on New Jersey. The CVI was validated by comparing index results to real-world COVID-19 data from New Jersey’s 21 counties and CDC’s SVI. The results present a stronger positive linear relationship between the CVI and the New Jersey COVID-19 mortality/population and infection/population than there is with the SVI. The results of this study indicate that Essex County has the highest CVI, and Hunterdon County has the lowest CVI. This is due to factors such as disparity in wealth, population density, minority status, and housing conditions, as well as other factors that were used to compose the CVI. The implications of this research will provide a critical tool for decision makers to utilize in allocating resources should another global pandemic occur. This CVI, developed through this research, can be used at the county, state, and global levels to help measure the vulnerability to future pandemics.

## 1. Introduction

With the world facing uncertainty in the midst of a global pandemic, many researchers and decision makers have turned to statistics in order to interpret data and understand the severity of the COVID-19 pandemic, which was brought upon by the coronavirus disease 2019. Facets of everyday life have seen a change in response to the COVID-19 pandemic, including changes seen from both societal and economic perspectives. With a USD 9 trillion global expense and millions of lives lost, everyday life has changed drastically since the COVID-19 outbreak was declared a global pandemic by the World Health Organization (WHO) in March of 2020 [1]. With a bleak future on the horizon, it became increasingly evident that there is a need to determine which areas and communities are most affected and, in turn, most vulnerable to the effects of the COVID-19 pandemic. Available resources, such as the Center for Disease Control’s (CDC) Social Vulnerability Index (SVI), are instrumental in predicting vulnerabilities within a community, as the utilization of precedents and historical data has never been more important.

The need for a COVID-19 Vulnerability Index (CVI) became evident due to the precedent set by the COVID-19 pandemic. With millions of people infected in the state of New Jersey alone, it became imperative that each county of New Jersey be assessed and given a statistically determined vulnerability evaluation in relation to the COVID-19 pandemic. The CDC’s SVI is a great asset for the people of the U.S. The CDC’s SVI provides important estimates on which communities will be more susceptible to ‘hazard events’ by compiling a variety of data from the U.S. Census and the American Community Survey. However, the CDC’s SVI does not necessarily consider the susceptibility of a community to a global pandemic, such as the COVID-19 pandemic. Even though the CDC’s SVI may not provide the most accurate data in terms of pandemic vulnerability, portions of it can still be utilized as assets when compiling data for a pandemic, specifically for a CVI. The CDC’s SVI was validated with real-world data; therefore, it is the best frame of reference for establishing the foundation for different types of vulnerable indices, such as the CVI [1,2].

The scope of the current literature is widespread, and there has not been a determination as to which methodology is the most accurate in determining a population that is susceptible to the COVID-19 pandemic. Numerous attempts were made to create an unbiased CVI utilizing different areas of study and resources. The primary concept highlights the different methods that are being implemented at different scales around the world. Amongst the varying methods of creating a CVI, there are several methods that seem to be the most relevant, such as the AHP (Analytic Hierarchy Process), generalized propensity scoring used in conjunction with time-varying data, and machine learning, and were applied to CVI generation [3,4,5,6]. Among the highly diverse areas of methodology, the one objective that they all share is to establish a better understanding of the COVID-19 pandemic and the susceptibility of different populations to it regardless of the methodology [5,6,7,8,9].

There are several steps that need to be taken to develop a comprehensive CVI. The CDC’s SVI, as mentioned previously, paved the way in terms of research and development for the generation of what an index of this magnitude should look like. Through this, they established a fundamental framework by which this research can follow. This includes the use of publicly available datasets for the counties of New Jersey, as well as theme development. The motivation behind this study is to generate public awareness for the counties of New Jersey that may have a more vulnerable population, as well as to generate an unbiased index for the state of New Jersey. The inspiration for this study came from the University of Maryland’s publication on COVID-19 and the relationship between racial inequalities [9]. There was mention of the creation of an index that mapped out contributing factors to COVID-19 vulnerability. The research conducted by the Surgo Foundation involved the supplemental integration of a community’s demographics into the consideration of a CVI [5]. Several studies have shown that the COVID-19 pandemic and other disasters disproportionately affect minority groups [10,11,12,13,14].

The overall objective of this research is to develop a COVID-19 Vulnerability Index (CVI) and use New Jersey counties to validate the index. This research has the following two main objectives:(1)To develop a CVI that will aid the preparedness for future global pandemics;(2)To incorporate New Jersey county data into the CVI to validate the index.

By achieving these objectives, this research builds on the established field of indices and provides a better understanding of how to determine which communities are more susceptible to the COVID-19 pandemic, as well as to future global pandemics.

## 2. Methodology

Through this research, a COVID-19 Vulnerability Index (CVI) was developed for the counties of New Jersey by utilizing the existing framework and methodology from those of the CDC’s SVI. Previous research studies, as well as publicly available datasets, were utilized in the development of the CVI [5,8,9,15]. As can be found through various aspects of the current literature and research, many COVID-19 vulnerability studies and indices seem to conform to the methodology established by the CDC’s SVI [3,4,5,16,17]. The CDC’s SVI has created a fundamental foundation for establishing an index related to public health. This is established by configuring themes and factors that are generalized and related to the index that is proposed. The key indices that were established by the current literature that are the most applicable to the development of a CVI are the CDC’s SVI and the Surgo Foundation’s COVID-19 Community Vulnerability Index (CCVI) [5]. Where the CDC’s SVI focuses more on the vulnerability aspect in relation to natural disasters, the Surgo Foundation’s CCVI focuses more on vulnerability to the COVID-19 pandemic [2,5].

The first primary reference to CVI construction in the literature is the Surgo Foundation’s CCVI. The Surgo Foundation’s CCVI aims to utilize the CDC’s SVI’s proven methodology of determining the susceptibility to natural disasters and rearranging the themes into those that suit the needs of a CVI as opposed to an SVI. The Surgo Foundation’s CCVI compares the counties of the U.S. to one another similarly to the SVI and rates them using the same methodology. The methodology that is followed in this study predominantly relates to the Surgo Foundation, because the CDC references it in their COVID-19 vulnerability section [1,2]. Where the CDC’s SVI has had the better part of a decade to determine whether its claims are accurate in relation to disaster events and how long communities took to recover, the Surgo Foundation’s CCVI was only developed recently and has been applied since the beginning of the COVID-19 pandemic. The Surgo Foundation has had a comparatively smaller window of time to validate their methodology due to the recent events surrounding the COVID-19 pandemic. The Surgo Foundation was one of the first to publish a complete CCVI. The Surgo Foundation’s work has given tremendous insight as to how the themes and the different variables regarding a CCVI should be organized and which methodology should be employed when dealing with such an undertaking.

The current CDC’s SVI establishes four themes (i.e., Socioeconomic Status, Household Composition and Disability, Minority Status and Language, and Housing Type and Transportation) to create a fundamental foundation for establishing an index related to public health [1,2]. The Surgo Foundation’s CCVI focuses on vulnerability to the COVID-19 pandemic by utilizing seven themes (i.e., Socioeconomic Status; Minority Status and Language; Housing Type and Transportation; Household Composition and Disability; Epidemiological Factors; Healthcare System Factors; High-Risk Work Environments; and Population Density) [5,15]. These two indices differ to represent a population’s vulnerability to natural disasters and the COVID-19 pandemic, respectively.

This study integrates the CDC’s SVI and the Surgo Foundation’s CCVI to develop a CVI which includes the COVID-19 pandemic and social vulnerabilities by considering eight themes, which can be seen in Figure 1.

These eight themes are comprised of numerous different factors, all contributing to the generation of a CVI. However, these eight themes and their composition may differ from other CVIs, such as the Surgo Foundation’s CCVI. Themes 1–7 (Table 1) were all inspired by the CDC’s SVI and the Surgo Foundation’s CCVI construction, respectively [5,15]. Theme 8 (Table 1) was generated due to the number of unique variables surrounding the state of New Jersey. For each theme, relevant factors were chosen, as seen in Table 1. These themes and their respective factors were selected and generated based on the current literature in the development of a CVI, as well as indicators that were determined to be directly related to the susceptibility of the COVID-19 pandemic. Table 1 predominantly stems its inspiration from the methodology established by the CDC, the Surgo Foundation, and the European Union [2,5,15]. This study is only composed of New Jersey counties and their respective data to develop the index.

Theme 1 (Socioeconomic Status) factors were predominantly derived from the CDC’s SVI [2]. Socioeconomic status is a crucial factor used to identify a CVI, as lower-income areas generally have less access to healthcare and/or assets to subsist on in the event of a pandemic [12,13,14].

Theme 2 (Minority Status and Language) is relevant in the formation of a CVI due to the fact that public health messages and different regulations may not translate very effectively for the percentage of the population that cannot speak or understand the English language very well. Minority status (i.e., all persons except white, non-Hispanics) is also considered when identifying the CVI due to the lack of facilities that minorities tend to have access to. Several studies have researched the impacts of natural disasters and other social phenomenon in order to better understand their effects on populations composed of minorities and how these people were disproportionately affected by the pandemic [12,13,14].

Theme 3 (Housing Composition/Type, Transportation, and Disability) is composed of numerous categories that relate to housing type and composition, as well as population statistics. More specifically, crowding, multi-unit structure, and group quarters data from the SVI were compiled and used, which is related to population density and directly relates to contact exposure. Other values that were considered for Theme 3 were the percentage(s) of the population that lived in mobile homes, were aged 17 years or younger, and/or did not have access to a personal vehicle. These factors were important in determining portions of the population living in less-than-ideal circumstances, as well as finances. Otherwise, this theme indicates what percentage of the community will be using some form of public transportation. These factors are all extremely important when considering the determination of a CVI. This is due to the fact that though population density may be a generalized average for a large area, these factors are much more specific in directly determining what percentage of the population is living in less-than-ideal conditions in relation to the pandemic. All of these factors contribute to the possibility of exposure to the virus and potential health risks associated with said living conditions/choices [18,19,20].

Theme 4 (Epidemiological Factors) consists of data (i.e., pre-existing health conditions, age, and lifestyle choices) gathered from the New Jersey Department of Health Portal [21]. The pre-existing categories that compose this theme are conditions such as cardiovascular health, respiratory health, immunocompromised factors, and age. For cardiovascular health, the following values were considered: heart disease, chronic obstructive pulmonary disease (COPD), diabetes, high cholesterol, stroke history, and obesity. These health factors increase the chance of mortality if exposed to COVID-19, as well as increase the likelihood of frequenting a hospital setting for treatment related to these conditions. Respiratory health is an issue of large concern when it comes to COVID-19. This is due to the fact that the virus heavily affects the respiratory system and can cause pneumonia, among other respiratory illnesses. For those with pre-existing respiratory conditions, this effect is catalyzed and can lead to high mortality rates. The factors that contribute to these factor areas are the percentage of the population that smokes cigarettes and those who have COPD. The immunocompromised factors consist of the following two factors: (1) per capita that is afflicted with HIV and (2) percentage of the population that is afflicted with cancer. These were both measured per 100,000 people. By having immune system deficiencies, the virus will most likely have a stronger effect and lead to higher mortality rates in those with said conditions. Another factor to consider is that New Jersey has the highest number of superfund sites in the U.S., most of which are active and contaminated with known human carcinogens or toxins [22]. Therefore, this value is rather important with respect to the New Jersey case study. There is a large role that pre-existing health factors play in the mortality rate of COVID-19. As mentioned, the more severe the pre-existing conditions, the higher the chance of mortality and/or severe health complications if one contracts COVID-19 [17].

Theme 5 (Healthcare System Factors) comprises numerous categories that represent the healthcare systems related to each county of New Jersey. The categories that comprise this theme are as follows: capacity, strength, and accessibility of healthcare systems. Beyond these categories, there are several factors, such as ICU (intensive care unit) beds per 100 k, hospital beds per 100 k, cost of medical care, and the number of primary physicians per 1000, where the 100 k and 1000 for these factors are a measure of population size. The number of emergency department visits and total hospital visits surged from March 2020 to January 2021 within the study domain [23]. This made it critical that those needing emergency care for treatment of COVID-19 or life-threatening illnesses have access to such facilities. By incorporating these categories in the CVI, it provides a picture of the various factors that come into play when assessing the effectiveness of the healthcare system(s) in each respective county. In any pandemic or disaster event, health system preparedness and strength are always extremely important factors for obvious reasons. In relation to COVID-19, certain counties may not have as much capacity and/or resilience to serve their respective communities. Unfortunately, this was the case for many patients afflicted with the virus. By understanding the healthcare infrastructure that is established in each county, it is possible to determine which counties may need state and/or federal assistance on the healthcare front. It is also important to note that counties that exceed capacity may have residents seeking medical attention in neighboring counties and even states.

Theme 6 (High-Risk Work Environments) comprises three categories that represent the percentage of the population living and/or working in high-risk environments with respect to COVID-19 [5]. This theme also comprises two factors, including prison population per 100,000 residents and long-term care residents per 100,000 residents of each county. These work and living environments are extremely high risk, which means that there is a high chance of COVID-19 transmission in these areas. By determining the percentage of the population living and/or working in these environments, we can gain a better understanding of what percentage of the population risks COVID-19 exposure on a daily basis simply from their work and/or living environment. Examples of this high-risk industry include dental hygienists, healthcare workers, social workers, etc. By having a high percentage of the population working in these professions, it is much more likely that a virus can spread and affect multiple families.

Theme 7 (Population Density) is one of the most important factors, which is why it is in itself a stand-alone value from county to county. Population density is one of the most important indicators of COVID-19 transmission, as well as an indicator of numerous other factors [24]. This is due in part to the nature of the virus, in that it is predominantly spread in three ways, which relate to breathing and bodily fluids. These routes of exposure are droplets which can be spread through various means, such as breathing, touching the face with contaminated hands, and having droplets land on the face from a sneeze and/or cough [25]. Theme 3 is related to population density due to factors such as multi-unit housing, crowding, and group quarters. Though these are independent in themselves, population density can provide a wider basis of the average population per area. As previously mentioned, population density is a more generalized value for how densely packed a county may be, whereas Theme 3 is a little more specific.

Theme 8 (Environmental Factors) is an appropriate addition to the index by this study. The factors that comprise this theme are air quality data that include the average daily PM 2.5, location of superfund sites, and vehicle traffic data. The average daily PM 2.5 relates to the amount of particulate matter in the air, which is a direct indicator of air quality. This is measured in μg/m^3^ and can have adverse health effects on entire communities if overexposed, and thus, higher COVID-19 infection/mortality rates [26,27]. Superfund sites were related to the amount of superfund sites per country and then divided by the total area of said county. This then determined the number of superfund sites per square mile. Considering that New Jersey has the highest number of superfund sites in the U.S., this factor is more than worth considering in the development of an index. Superfund sites are still being discovered and remediated to this day as more and more manufacturing processes are discovered. Vehicle traffic relates to the amount of traffic that passes through a county’s major roads. It is well known that vehicle emissions include BTEX (Benzene, Toluene, Ethylbenzene, and Xylene) compounds, which are known carcinogens and toxins. Vehicle travel also relates to the amount of traffic a county might experience on a regular basis. Considering the avenues by which COVID-19 spreads, high-vehicle traffic could relate to higher transmission rates.

Once general themes that are relevant to the research are established, the factors of these themes can then populate. The selection of these factors is determined by whether the data are publicly accessible, published/verified, and representative of the index construction (i.e., related factors to themes and overall vulnerability). These themes and their respective factors can be applied to any area of study.

This study develops four steps to define CVI values. First, relevant data are collected from a credible source that will benefit the CVI and will represent a factor in a theme. Secondly, the data are normalized by population or divided by 1 in order to keep the scaling consistent. The normalized data are then processed through the ‘RANK.EQ’ function, giving values from 0–1 for every given factor. Subsequently, the factors are summed together and averaged, which yields each theme value. Thirdly, each county’s theme values are summed and averaged, yielding a CVI value using Equation (1) below. Finally, each CVI value can then be processed by the ‘RANK.EQ’ function in order to understand which county has the highest and lowest CVI value with respect to every county. Equation (1) represents the CVI theme summation once the individual factors are calculated. Factors can be modified using the weighted factors in Equation (1).
(1)$$\mathrm{CVI}=\frac{\sum_{i=1}^{m}\left[\frac{\sum_{i=1}^{n}\left(x_{i}*w_{i}\right)}{\sum_{i=1}^{n}(n)}\right]}{m}$$
where *x* is factor value; *w* is weight; *n* is number of factors; and *m* is number of themes.

This study considered 21 counties in New Jersey as the study area and calculated CVI values for each county. Data related to each factor and theme were collected from different data sources, as seen in Table 2.

## 3. Results

Table 3 represents each county of New Jersey and the respective theme values, CVI values, and ‘RANKEQ’ CVI values. Themes 1–8 represent the CVI theme values before they are summed and averaged into a final CVI value, which can then be seen in the column labeled CVI in Table 3. The theme value(s) of each theme are in relation to the factors and their subsequent normalized value. The individual factor value(s) were standardized by population where applicable, and then organized on a scale from 0–1 using the ‘PERCENTRANK’ function in Excel. By acknowledging the different individual theme values, one can make delineations as to which counties may be more susceptible in certain areas/themes as opposed to others. In the column labeled RANK EQ. CVI in Table 3, the values that are calculated from the CVI are then organized from least to greatest in order to readily determine which counties have the highest respective CVI value(s). This study considered either the RANK EQ. CVI values or the CVI values to further investigate the results and validate the methodology proposed.

Figure 2 represents the data which were normalized using the RANK.EQ function. This study considered the SVI values determined by the CDC from March 2020 to December 2020 [8]. Each county was ranked using a color-coded scale spanning from the following: Low (0.0–0.25), Moderate (0.251–0.50), High (0.51–0.75), and Very High (0.751–1). There are many similarities between the different maps, as can be seen with the northeastern counties. This predominantly includes the Essex, Passaic, Hudson, and Union counties. It is rather interesting that although the data are derived from completely different areas of research, there are trends emerging. This data helps validate the themes and factors that were instilled in the CVI match the real-world comparisons of mortality and infection data from the COVID-19 pandemic.

Figure 3 represents the CVI developed by this study in comparison to the SVI values determined by the CDC through linear regression. The R^2^ value was determined to be 0.5312. This means that there was a slight amount of variance between the independent and dependent variables when comparing the CVI data with those of the CDC’s SVI, establishing a positive linear relationship. As seen in Table 4, the comparison between the CVI and SVI yields the R^2^ value of 0.713. With such a strong positive linear relationship and by considering methodologies established by authors in previous years [5,15], it was determined that the best way to represent the data as unbiased is by using weighting factors. With the proper arrangement of factors and themes, the data can speak for itself in the New Jersey case study. The weighting factors that were applied to the calculation of the CVI for this study were equal with respect to each theme. This can be seen in Equation (1) by applying a weight of 1 where the weight variables are considered.

Figure 4a compares the CVI values calculated in this study to COVID-19 mortality observed in New Jersey from March 2020 to December 2020, while Figure 4b compares the SVI values determined by the CDC to COVID-19 mortality observed in New Jersey during the same time period. As seen in Figure 4a,b, the CVI would be an accurate way to represent COVID-19 mortality. As can be seen in Figure 4a, the R^2^ value is 0.655, which is much greater than the R^2^ value of 0.3734 seen in Figure 4b. This indicates that the linear relationship between the CVI and COVID-19 mortality data is stronger than that of the SVI and COVID-19 mortality data. This means that the CVI can represent the number of COVID-19 deaths more accurately than the SVI.

Figure 5a compares the CVI values calculated in this study to COVID-19 infections observed in New Jersey from March 2020 to December 2020, while Figure 5b compares the SVI values determined by the CDC to COVID-19 infections observed in New Jersey during the same time period. As seen in Figure 5a,b, the CVI would be an accurate way to represent COVID-19 infections. As can be seen in Figure 5a, the R^2^ value is 0.6688, which is much greater than the R^2^ value of 0.4093 seen in Figure 5b. This also indicates that the linear relationship between the CVI and COVID-19 infection data is stronger than that of the SVI and COVID-19 infection data. Based on Figure 4 and Figure 5, this study concludes that the CVI can represent the number of COVID-19 deaths and infections more accurately than the SVI.

Table 4 represents the linear regression values between the following: CVI vs. SVI, CVI vs. COVID-19 infections/population, and CVI vs. COVID-19 deaths/population. In Table 3, there are several parameters that were measured when conducting the linear regression analysis. The generalized linear regression function (also known as least squares) seen in Equation (2) was applied to conduct a linear regression analysis. This study used the Excel linear regression function. This study also investigated the R^2^ (coefficient of determination), R (Pearson’s correlation coefficient), and Significance F (*p*-value).

Pearson’s correlation coefficient, which measures the strength of the linear relationship between both the independent and dependent variables, was calculated using Equation (2a). The R value directly relates to the strength between two variables in a linear trend. The higher the R value, the stronger the linear trend and, subsequently, the higher the R^2^ value. The R^2^ value is a measure of variance that can be seen through an analysis of the independent and dependent variables. This effectively represents how much the variables relate to one another numerically. The higher the R^2^ value, the less variance there is between the independent and dependent variable data.

The Significance *F* (*p*-value) can be calculated using Equation (2b). The Significance *F* value represents the *p*-value for the overall linear regression graph. This value should not exceed 0.05 if there is to be a case made for the alternative hypothesis (linear relationships) to be stated as true. As seen in Table 4, there is a high correlation between the CVI and real-world data. Thus, the CVI values that were generated are concurrent with the real-world data that were collected regarding COVID-19 deaths and infections observed in each county of New Jersey.
Y_i_ = f (X_i_, β) + e_i_(2)
where Y_i_ is the dependent variable (real-world COVID-19 data), X_i_ is the independent variable (CVI values), β is the unknown parameters, and e_i_ = error items.
(2a)$$R=\frac{\Sigma \left(x_{i}-\overset{-}{x}\right)\left(y_{i}-\overset{-}{y}\right)}{\sqrt{\Sigma }{\left(x_{i}-\overset{-}{x}\right)}^{2}\;\Sigma {\left(y_{i}-\overset{-}{y}\right)}^{2}}$$
(2b)$$F=\frac{\frac{R^{2}}{k}}{\frac{1-R^{2}}{N-k-1}}$$
where *x_i_* is the independent variable values, $\overset{-}{x}$ is the mean of the independent variable, *y_i_* is the dependent variable values, and $\overset{-}{y}$ is the mean of the dependent variable.

Table 4 represents the linear regression values that are visualized in Figure 3, Figure 4 and Figure 5. As stated previously, there is a stronger positive linear relationship between the CVI and the New Jersey COVID-19 death/population and infection/population than there is with the SVI. This can be seen through the Multiple *R*, *R*^2^, and Significance *F* values in Table 4.

## 4. Discussion

By analyzing the CVI values resulting from this study, the successful implementation of eight themes and 39 different factors into a CVI was accomplished, which will aid the determination of the vulnerability of a community to the COVID-19 pandemic, as well as future possible global pandemics. Other emerging indices, such as the COVID-19 Stringency Index [35], utilize nine indicators, including school closures, travel restrictions, the cancellation of public events, etc., which are all actions that are reflected after outbreaks have occurred, causing federal, state, and municipal agencies to act. This differs from the CVI developed in this study, because the CVI aims to estimate the vulnerability of specific populations before these closures happen and give a generalized idea of which counties will need more financial support and resources in the event of a pandemic.

The novelty of the CVI and the way that it differentiates itself from other established indices are characterized by how it strictly utilizes county datasets to generate an index for an individual state. By focusing on one state, the data are more representative and do not become clouded by other states as mentioned by ‘flagging factors’ in the SVI. The methodology including various factors to determine the vulnerability to COVID-19 was validated based on New Jersey data, and this approach can be readily employed globally to comprehend the vulnerability of populations to pandemics, such as COVID-19.

However, there are still opportunities for future studies to expand upon the research conducted in this study. The weighting of the factors was not directly applied, as the importance of each factor is up to SMEs (subject matter experts) [5]. With that being said, every factor in each theme received a weight of 1, and thus, were all equivalent before the CVI calculations began. In this research, the only way that passive weight was applied to the CVI was by only including one factor in the population density theme due to the importance of COVID-19 being spread through human interaction. This follows the same principles as the Surgo Foundation’s SVI [5]. There was effectively no direct method of weighting applied to the factors. For this reason, there were limited options regarding sensitivity and uncertainty analyses. In the future, it may be advisable to include a methodology of weighing different case studies and areas that this research may be applied to. For the purposes of this study, it was seen as unnecessary to weigh values considering the results as discussed. For other areas of study that may see a prevalence of different factors, a weighted scale will help to determine COVID-19 vulnerability more accurately.

The other limitations of this study are the datasets that are available to the public, as well as the accuracy of said datasets. Otherwise, this study is solely focused on developing a CVI for the counties of New Jersey based on population, health, and environmental metrics. In the future, it would be remarkable if municipality data could be incorporated into the CVI, and therefore, municipalities, counties, and states would have a greater understanding of how COVID-19 and other future pandemics need to be addressed on micro and macro scales within populations.

As seen in the current literature [2,4,5,6,7], there is a magnitude of angles that research can follow when establishing an index in this field. With that being said, the index could include more values that directly correlate with COVID-19 susceptibilities, such as vaccine distribution and several other factors that may be relevant. Otherwise, further research should focus on developing a time-varying index.

Death, infection, and vaccination rates are always changing through the event of a pandemic, and thus, the vulnerability of certain populations may change in the short or long term. This is a much more complex index to develop, but it would be ideal for determining the long-term impacts of a future global pandemic. This CVI, developed through this research, can be used at the county, state, and global levels to help measure the vulnerability of populations to future pandemics.

## 5. Conclusions

The methodology and factors used for developing the CVI that was proposed by this study represents the effects of the COVID-19 pandemic better than the CDC’s SVI, as investigated using New Jersey data. The importance of accounting for the eight themes that comprise the CVI (i.e., Socioeconomic Status, Minority Status and Language, Housing Type and Transportation, Epidemiological Factors and Disability, Health System Capacity, High-Risk Work Environments, Population Density, and Environmental Factors) are represented in the results; these themes all play an important role in accounting for populations that may be susceptible to the COVID-19 pandemic. The results from the CVI provide further insight into identifying which communities and counties are more susceptible to COVID-19. The findings from this research suggest that the manipulation of themes and factors will aid the accuracy of the CVI in the future. The addition of municipalities to the CVI and the use of theme and factor weights will determine which communities are more at risk to COVID-19 and future pandemics with a greater accuracy.

## Figures and Tables

**Figure 1 ijerph-20-06312-f001:**
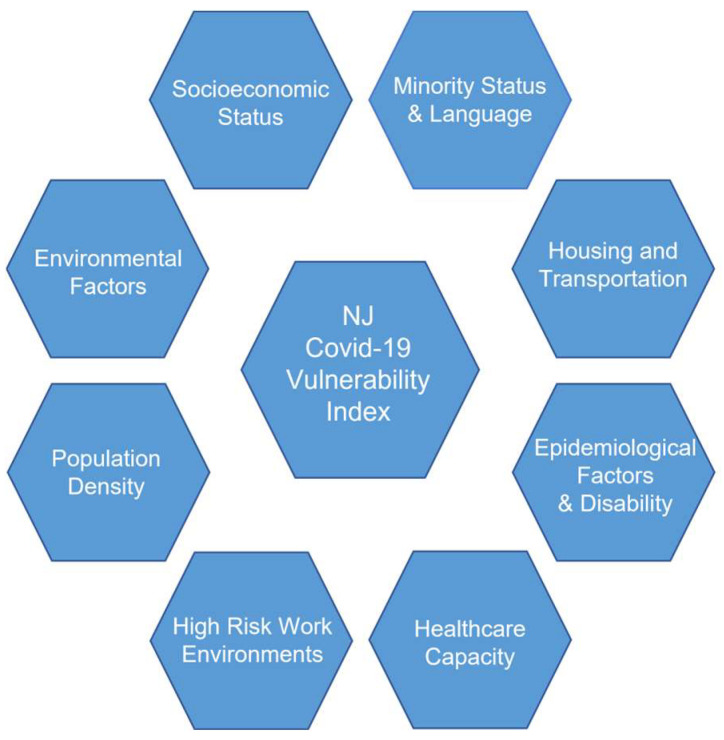
Theme structure developed for the creation of a COVID-19 Vulnerability Index (CVI) for the counties and residents of New Jersey.

**Figure 2 ijerph-20-06312-f002:**
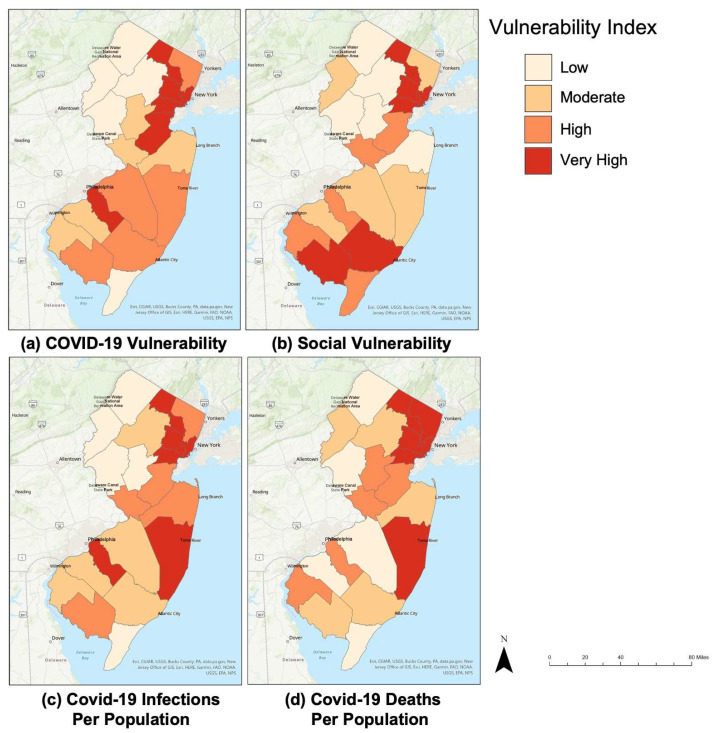
(**a**) CVI values determined by this study for the 21 counties of New Jersey, (**b**) SVI values developed by the CDC, (**c**) COVID-19 infections/population, and (**d**) COVID-19 deaths/population for the 21 counties of New Jersey.

**Figure 3 ijerph-20-06312-f003:**
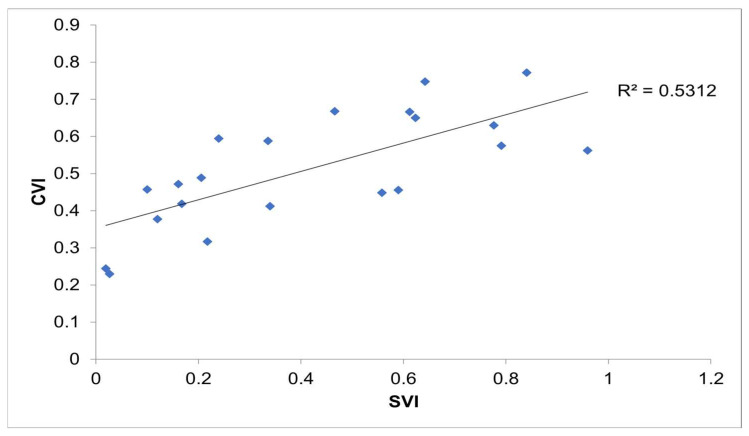
Comparison between the CVI values calculated in this study and the SVI values developed by the CDC.

**Figure 4 ijerph-20-06312-f004:**
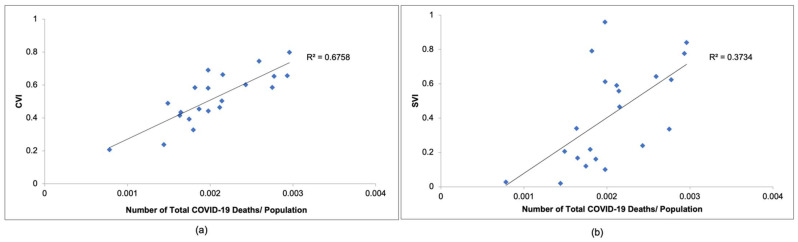
Comparison between (**a**) CVI and (**b**) SVI indices and number of COVID-19 deaths/total population.

**Figure 5 ijerph-20-06312-f005:**
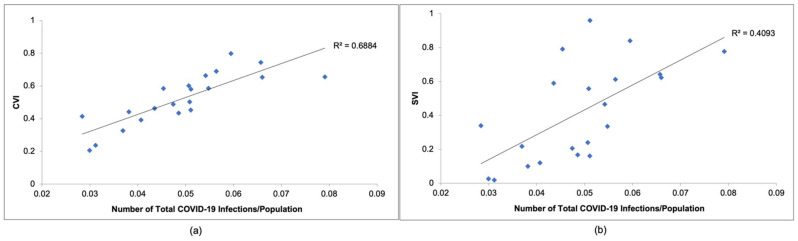
Comparison between (**a**) CVI and (**b**) SVI indices and number of COVID-19 infections/total population.

**Table 1 ijerph-20-06312-t001:** The eight themes and corresponding factors that comprise the COVID-19 Vulnerability Index (CVI) developed for the counties and residents of New Jersey.

Theme	Theme Title	Categories	Factors
Theme 1	Socioeconomic Status	Socioeconomic Status	Below poverty
			Unemployed
			Income
			No high school diploma
			Uninsured
Theme 2	Minority Status and Language	Minority Status and Language	Minority
			Aged 5+ years and speak English less than well
Theme 3	Housing Type, Transportation, Household Composition, and Disability	Housing Type	Multi-unit structures
			Crowding
			Group quarters
			Mobile homes
		Household Composition	Aged 17 years or younger
			Single parent household
		Transportation	No vehicle
		Disability	Disability
Theme 4	Epidemiological Factors	Cardiovascular Conditions	High cholesterol
			Stroke
			Heart disease
		Respiratory Conditions	COPD
			Cigarettes
			Asthma
		Immunocompromised	Cancer per 100 k
			HIV per 100 k
		Obesity	BMI > 30
		Diabetes	Diabetes
		Elderly	Aged 65+ years
Theme 5	Healthcare System Factors	Health System Capacity	ICU beds per 100 k
			Hospital beds per 100 k
		Health System Strength	PQI (Prevention Quality Indicator) per population
			Cost of medical care
		Healthcare Accessibility	Pharmacies and drug stores per 100 k
			Primary care physician per 1000
Theme 6	High-Risk Work Environments	Percent of Population Working/Living in High Infection Risk Environment	Long-term care residents per 100 k
			Prison population per 100 k
			Population employed in high-risk industry per 100 k
Theme 7	Population Density	Population Density	Populus per sq. mile
Theme 8	Environmental Factors	Average Daily PPM ‘16	Average daily PPM ‘17
		Number of Superfund Sites per County	Number of superfund sites per county
		Vehicle Volume	Vehicle volume

**Table 2 ijerph-20-06312-t002:** Themes and sources of data for factors.

Theme	Data Sources
Theme 1	U.S. Census [28]
Theme 2	U.S. Census [28]
Theme 3	U.S. Census [28]
Theme 4	U.S. Census, NJSHAD, and NJDOH [21,28,29]
Theme 5	KHN, KFF, and AHD [18,30,31]
Theme 6	HIFLD and Vera Institute [32,33]
Theme 7	U.S. Census [28]
Theme 8	EPA and CHR&R [22,34]

**Table 3 ijerph-20-06312-t003:** CVI values calculated for each of the 21 counties of the study area of New Jersey.

County	Theme 1	Theme 2	Theme 3	Theme 4	Theme 5	Theme 6	Theme 7	Theme 8	CVI	RANK EQ. CVI
Atlantic	0.9524	0.6905	0.7347	0.6151	0.4694	0.6508	0.3333	0.3810	0.6034	0.6190
Bergen	0.4667	0.6667	0.4150	0.3532	0.5646	0.6667	0.8571	0.8571	0.6059	0.6667
Burlington	0.4095	0.3810	0.4490	0.6746	0.5102	0.5397	0.3810	0.5873	0.4915	0.4762
Camden	0.8190	0.6429	0.6395	0.6508	0.4150	0.7460	0.7143	0.8413	0.6836	0.8571
Cape May	0.5714	0.1667	0.5646	0.7857	0.4558	0.5079	0.2857	0.2063	0.4430	0.2857
Cumberland	0.9619	0.7619	0.7687	0.7262	0.4762	0.4603	0.2381	0.2857	0.5849	0.5714
Essex	1.0381	0.8571	0.7755	0.5159	0.4014	0.7619	0.9524	0.8413	0.7679	1.0000
Gloucester	0.4762	0.1905	0.4218	0.5317	0.7211	0.3810	0.4286	0.4603	0.4514	0.3333
Hudson	0.9524	0.9762	0.6122	0.4722	0.3946	0.4762	1.0000	0.9206	0.7256	0.9524
Hunterdon	0.1143	0.0952	0.2585	0.3532	0.6122	0.1111	0.1429	0.3333	0.2526	0.0476
Mercer	0.3143	0.3571	0.2449	0.3690	0.4286	0.5714	0.7619	0.3968	0.4305	0.2381
Middlesex	0.8857	0.9286	0.7347	0.4722	0.5170	0.6667	0.6190	0.6667	0.6863	0.9048
Monmouth	0.5429	0.5238	0.5238	0.3373	0.4694	0.5714	0.5238	0.4286	0.4901	0.4286
Morris	0.1524	0.4048	0.3197	0.4484	0.5510	0.3810	0.4762	0.3651	0.3873	0.1905
Ocean	0.8286	0.3810	0.6939	0.5992	0.5306	0.8413	0.6667	0.5556	0.6371	0.7143
Passaic	0.9905	0.8810	0.7007	0.4921	0.4558	0.3810	0.8095	0.4444	0.6444	0.7619
Salem	0.7810	0.3571	0.5646	0.7976	0.5034	0.4921	0.0476	0.3333	0.4846	0.3810
Somerset	0.3238	0.5476	0.3265	0.4286	0.7483	0.5714	0.5714	0.6190	0.5171	0.5238
Sussex	0.2667	0.0476	0.2313	0.4762	0.6259	0.2540	0.0952	0.2222	0.2774	0.0952
Union	0.8190	0.8810	0.5986	0.3968	0.4898	0.4286	0.9048	0.8095	0.6660	0.8095
Warren	0.5333	0.2619	0.3946	0.3849	0.5646	0.2222	0.1905	0.3333	0.3607	0.1429

**Table 4 ijerph-20-06312-t004:** Comparisons between values for the CVI developed in this study and the CDC’s SVI, CVI and infections/population, and CVI and death/population for the counties of New Jersey.

Regression Statistics	CVI vs. SVI	CVI vs. COVID-19 Death/Population	CVI vs. COVID-19 Infections/Population
Multiple R	0.764	0.824	0.829
R^2^	0.584	0.676	0.688
ANOVA: Significance F	5.49 × 10^−5^	4.85 × 10^−6^	3.29 × 10^−6^

## Data Availability

Data are available upon request to the corresponding authors.

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
