# Peer review of "Development of a COVID-19 Vulnerability Index (CVI) for the Counties and Residents of New Jersey, USA"

_ijerph, 2023, doi:10.3390/ijerph20136312_

Round 1

Reviewer 1 Report

Dear authors:

A-      The provided abstract lacks specific details and context, resulting in a somewhat vague and general overview. Here are some critical comments:

  1. Lack of quantitative information: The summary mentions the development of a COVID-19 Vulnerability Index (CVI) based on 41 factors, but it fails to provide specific details about these factors or their relative weights. Without this information, it is challenging to assess the accuracy and effectiveness of the CVI.
  2. Limited validation: The summary states that the CVI was validated using real COVID-19 data from New Jersey, but it does not elaborate on the validation methodology or the extent of the validation process. Without transparency and rigorous validation procedures, the reliability and usefulness of the CVI are questionable.
  3. Missing comparison with the CDC's SVI: The summary mentions the CDC's Social Vulnerability Index (SVI) as an existing tool but fails to explain how the CVI differs or improves upon it. Providing a comparative analysis would have helped to highlight the novelty and value of the CVI.
  4. Lack of broader implications: While the summary mentions that the CVI will aid decision-makers in resource allocation during future pandemics, it does not elaborate on the broader implications or potential applications of this research. Exploring the potential impact of the CVI on public health strategies or policy-making would have added depth to the summary.

Overall, the summary would benefit from additional details, clarity, and a more critical analysis of the research's limitations and implications. The main results and conclusions of the study should be included in the abstract.

B-      The objectives of this study were to develop a COVID-19 Vulnerability Index (CVI) for future pandemic preparedness and validate it using New Jersey county data. However, the summary lacks specific details on how these objectives were addressed in the discussion and conclusion sections, making it difficult to evaluate the study's findings and outcomes accurately.

C-      It is unclear how the R2 value was obtained. Including a scatter plot or correlation graph would have been more informative and visually depict the relationship being measured. This would provide a clearer understanding of the statistical analysis and strengthen the presentation of the results.

Author Response

Thank you for your feedback. Please find attached: our responses.  

Reviewer 2 Report

Thank you for this interesting article proposing a new COVID-19 vulnerability index that appears to better represent population vulnerability to Covid-19 pandemic. Here are two suggestions to improve the understanding and potential use of the index: Provide, as supplement data, a spreadsheet with an example of the variables and data used to calculate the index.  Expand the Discussion section to explore possible limitations of the study and applicability of the index, such as: availability and quality of data, the need for weighting, inclusion of additional health-related factors, and the risk of a bias in the estimation.

Author Response

(The authors gave the same response as above.)

Reviewer 3 Report

I think it's important to explore other indices, and I commend your group for revisiting some COVID-19 era statistics and developing your own index. It's no easy feat and can invite a lot of criticism. For the most part, I think you've covered the important aspects and the paper can contribute something unique. Please see below for comments.

1. Figure 1. I wonder if you could consider removing Figure 1 since you already describe it in Lines 140-145?

2. Figure 2. Could you please change Figure 2a, 2c, and 2d captions from "Covid" to "COVID-19"?

3. Line 332-340. When you mention "linear relationship" you could even go beyond and state they are positive linear relationships.

4. Figures 3-5. I would strongly recommend moving these to the supplementary material since you're already reporting the R2 values in the text.

5. The results express your knowledge well, but I think you could be better off omitting some of the equations and processes in your paper (e.g., regression equation, correlation coefficient, F statistic). If you really want to, you could move it to the Appendix/supplementary material.

6. Table 4. Instead of "R Square" and "E-4" perhaps use the notation where you superscript those values so it's easier to read (i.e., R2).

7. Discussion. I'm coming from an epidemiological background so I think by its very nature our preferences for structuring manuscripts will differ and I'll try to be careful with my criticisms. From my perspective, your discussion section fell really rushed. Could you perhaps touch on the following in your discussion: (1) how your index compares to other existing indices (I understand the CVI is unique, but at least try to talk about some others like the stringency index and hazard index), (ii) scaling this up to beyond New Jersey, and (iii) add a limitations section for your paper to discuss shortcomings of the study and CVI. More importantly, you should be citing literature in your discussion section because you're essentially interpreting your study findings and making sense of it in the broader context using the available published knowledge.

Author Response

(The authors gave the same response as above.)
